# Insights into the molecular mechanism of yellow cuticle coloration by a chitin-binding carotenoprotein in gregarious locusts
Nikita A. Egorkin[1,2], Eva E. Dominnik[1,3], Eugene G. Maksimov [2] & Nikolai N. Sluchanko [1]✉

Carotenoids are hydrophobic pigments binding to diverse carotenoproteins, many of which remain unexplored. Focusing on yellow gregarious locusts accumulating cuticular carotenoids, here we use engineered *Escherichia coli* cells to reconstitute a functional water-soluble β-carotene-binding protein, BBP. HPLC and Raman spectroscopy confirmed that recombinant BBP avidly binds β-carotene, inducing the unusual vibronic structure of its absorbance spectrum, just like native BBP extracted from the locust cuticles. Bound to recombinant BBP, β-carotene exhibits pronounced circular dichroism and allows BBP to withstand heating ($T_{0.5}$ = 68 °C), detergents and pH variations. Using bacteria producing distinct xanthophylls we demonstrate that, while β-carotene is the preferred carotenoid, BBP can also extract from membranes ketocarotenoids and, very poorly, hydroxycarotenoids. We show that BBP-carotenoid complex reversibly binds to chitin, but not to chitosan, implying the role for chitin acetyl groups in cuticular BBP deposition. Reconstructing such locust coloration mechanism in vitro paves the way for structural studies and BBP applications.

Carotenoids are a diverse group of isoprenoid chemicals which serve as antennas, antioxidants, precursors of signal molecules, and colorants[1]. The hydrophobic nature of carotenoids dictates the necessity of their adaptation to the aqueous media via binding to water-soluble carotenoproteins[2,3]. Among few carotenoid-binding proteins studied is the Orange Carotenoid Protein (OCP), a photoswitching protein binding a ketocarotenoid—3'-hydroxyechinenone, echinenone (ECH) or canthaxanthin (CAN)—and conferring high-light tolerance to OCP-encoding cyanobacteria[4–7]. Some green microalgae respond to photooxidative stress by accumulation of the unrelated astaxanthin-binding protein AstaP[8], which binds various oxygenated carotenoids (xanthophylls) and, much less efficiently, β-carotene[9,10]. The exact function of AstaP remains largely unknown.

Many species use carotenoproteins for the body coloration. A well-studied example of colorant carotenoproteins is β-crustacyanin in crustaceans[11]. Binding of astaxanthin from lobster diet to β-crustacyanin entails the so-called bathochromic shift, from 475 nm in acetone to ~580 nm, imparting the characteristic blue-violet coloration to the lobster shells[12,13]. Another body-coloring carotenoprotein with a considerable bathochromic shift and a distinct ependymin fold unrelated to β-crustacyanin has recently been discovered in sea sponge *Haliclona sp.*; this protein binds dietary xanthophylls astaxanthin and mytiloxanthin[14].

Insect body coloration can be determined by diverse mechanisms including changes of microstructures, refractive index or pigment content of the integument. Insect pigments also vary and comprise ommochromes, melanins, pteridines, purines, bile pigments, and carotenoids[15,16]. A rare example is a pea aphid *Acyrthosiphon pisum*, which acquires yellow and red body coloration via accumulation of atypical carotenoids dehydro-γ,ψ-carotene and torulene produced by aphid itself; this is achieved owing to the horizontal transfer of gene cluster for carotenoid synthesis from fungi[17]. Remarkably, the same carotenoids are found in the body integument of ladybirds *Coccinella septempunctata*[18], predatory beetles naturally fed on aphids. To the best of our knowledge, no carotenoprotein has been identified in these species to date, but this certainly deserves attention in the future research.

Another example is much more studied. The yellow color of silkworm cocoons is determined by lutein and some other carotenoids ingested by

¹A.N. Bach Institute of Biochemistry, Federal Research Centre of Biotechnology of the Russian Academy of Sciences, Moscow, Russia. ²M.V. Lomonosov Moscow State University, Faculty of Biology, Moscow, Russia. ³M.V. Lomonosov Moscow State University, Faculty of Chemistry, Moscow, Russia. ✉e-mail: nikolai.sluchanko@mail.ru

silkworms with mulberry leaves they feed on. Lutein is accumulated in the silk gland and then deposited in the silk due to the activity of an intracellular transport protein from the steroidogenic acute regulatory transfer (START) family, BmCBP (*Bombyx mori* Carotenoid-Binding Protein)[19,20]. Besides its native ligand lutein, BmCBP binds various xanthophylls including the lutein's isomer zeaxanthin (ZEA), as well as traces of carotenes[21–23]. In vitro, BmCBP can extract xanthophylls from crude herbal extracts and transfer them to mammalian cells to counteract the oxidative stress[22,24]. Upon engineering carotenoproteins can be optimized for the use as antioxidant delivery modules in biomedical and biotechnological applications. However, all studied and sequenced carotenoproteins have a very limited capacity to extract and solubilize β-carotene, a ubiquitous carotenoid particularly valuable to human health.

A very peculiar case of insect coloration is represented by locusts—important and abundant pests that quickly eat and breed, switching from solitary forms to swarms and seriously threatening agriculture by rapid extermination of plantations. Therefore, locust physiology is in focus of intense research aimed at eventually controlling the locust gregarization and preventing substantial economic losses[25]. Locusts are known for their polyphenism associated with the ability to change body color at different phases of their development and lifestyle, from green to black/brown and bright yellow. They feed on various plant sources and therefore consume a lot of carotenoids[26,27]. While roughly similar carotenoid content is found in females and males of gregarious locusts, only males acquire bright yellow color in high-density populations, i.e., in the gregarious phase, to avoid harassment from other mature males[26–29]. This yellow coloration, also known to be induced in nymphs at high temperatures, is stimulated in males by juvenile hormone and probably some sex hormone(s), whereas melanization is controlled by the special neuropeptide corazonin[30–33]. Yellowing of gregarious locusts, such as *Schistocerca gregaria* and *Locusta migratoria*, results from the male-specific expression of the special carotenoprotein found in cuticles and epidermis[27–30,34,35]. Recently it was found that this is achieved by activation of protein kinase C alpha in response to crowding: PKCα phosphorylates the activation transcription factor 2 at Ser327 to promote its binding to the yellow protein promoter and induce overexpression[36]. Treatment of the yellow cuticles of gregarious locusts with 6 M urea enabled isolation of a protein-bound carotenoid chromophore[34]. This cuticular *yellow protein* was sequenced and its ability to bind β-carotene was proposed[27,29,34]. However, such tentative β-carotene-binding protein (BBP, in other works named as yellow protein, yellow protein takeout (YPT), or β-carotene-binding protein (βCBP)) was not characterized further. BBP sequence is homologous to proteins of the takeout superfamily that play a role in juvenile hormone transportation and have an α/β-wrap, the so-called TULIP fold[28,29,36–38]. The ability of BBP to bind and transport juvenile hormone or carotenoids has not yet been studied directly. Two recent works have attempted modeling of BBP structure but proposed contradicting mechanisms of β-carotene binding[28,36]. Another recent study[37] reported that recombinant GST-tagged BBP and its homolog ALTO (albino-related takeout) can bind lutein, but not β-carotene or astaxanthin, which brought a serious controversy and required independent studies with stringent controls.

To fill these gaps and eliminate controversies, here we focused on comparison of the native BBP extracted from the yellow locust cuticles and its recombinant variant produced in *Escherichia coli* cells synthesizing carotenoids. The recombinant BBP was successfully reconstituted in a soluble form, purified and analyzed by spectrochromatography, absorbance, circular dichroism, and Raman spectroscopies. This confirmed the equivalence of recombinant and native yellow protein and revealed its remarkable stability to heat treatment, urea, detergents, and pH variations. We demonstrated the preferences of BBP for β-carotene over xanthophylls and tested the ability of BBP:β-carotene complex to interact with chitin and chitosan. Our study provides the detailed characterization of BBP and thereby complements previous works in explaining the remarkable molecular mechanism of locust cuticle coloration. Our work also paves the way to structural studies and potential applications of this unique β-carotene-binding protein.

## Results

### Locust cuticles contain extractable yellow protein and retain pigmentation for long

We first questioned whether the yellow locust cuticle pigmentation is associated with a water-soluble protein that could be extracted and analyzed, to reveal for how long the pigmentation is retained in particular. To this end, we found two male locust specimens markedly differing by their age: one was kept in an entomology collection since 1959 and another one was cultivated until its natural death a few days before the experiment and had a bright yellow body color (Fig. 1a). Each specimen was dissected, fragments of its cuticles were rubbed and underwent ultrasonic treatment in a buffer containing no denaturants. Clarified by centrifugation, the cuticle residuals largely lost color, whereas the obtained extracts were yellow, indicating solubilization of the pigment (Fig. 1b), in agreement with earlier reports[34]. We performed acetone extraction and HPLC analysis, which confirmed that the yellow locust sample contained almost exclusively β-carotene in both the total and water-soluble extracts (Supplementary Fig. 1). We could not extract detectable amounts of water-soluble carotenoid forms from the old locust cuticle, and the yellowish color of the corresponding supernatant had a different tint (Fig. 1b). This may indicate that the carotenoid as well as protein envelope underwent some sort of sclerotization process and became immobilized to the cuticle during long storage.

To directly probe β-carotene signatures in situ—in the cuticles of the two locust specimens—and compare them with those of free β-carotene in n-hexane, we applied Raman spectroscopy (Fig. 1c). The Raman frequencies of the ν1 (1525 cm$^{-1}$), ν2 (1157 cm$^{-1}$), ν3 (1006 cm$^{-1}$), and ν4 (960 cm$^{-1}$) bands for β-carotene in n-hexane very well corresponded to those reported earlier[39]. The two analyzed locust cuticle spectra were remarkably similar to the Raman spectrum of β-carotene (Fig. 1c), yet the spectrum acquired from the 64-year-old male locust cuticle had a ν1 band, which is associated with C=C bond oscillation, shifted from ~1525 to 1511 cm$^{-1}$. It also had small features at ~957 cm$^{-1}$ reflecting the hydrogen-out-of-plane wagging modes[39], which are usually the most pronounced for carotenoids conformationally constrained by complexation with proteins[40,41]. Together these two signatures probably indicate that during the long-term storage of this sample the carotenoid adopted a more relaxed configuration upon sclerotization, which prevented direct extraction of the pigment by buffer. In contrast, the Raman spectrum of the yellow locust cuticle had a clear band at ~957 cm$^{-1}$ and a small but distinct shift of the ν1 band, together strongly suggesting the presence of a carotenoid-binding protein (Fig. 1c). Indeed, loading of the soluble yellow extract obtained from the locust cuticles on size-exclusion chromatography (SEC) yielded some aggregation and a major symmetric peak with an apparent $M_w$ of 50–55 kDa (Fig. 1d). The absorbance spectrum within this peak had a large Vis/UV absorbance ratio of ~4 and a pronounced vibronic structure in the visible region with three maxima at 428, 452, and 481 nm, i.e., >10 nm blue-shifted relative to the recently described *Bombyx mori* carotenoprotein complexed with ZEA, BmCBP(ZEA); the fine structure of the former was even more pronounced (Fig. 1e)[21,40]. This directly confirmed that the observed yellow color is determined by the β-carotene-binding protein, whose specific complex with β-carotene we term here BBP(βCar). In accord, the Raman signatures of BBP(βCar) were very similar to those of the leg sample of the yellow locust, including the pronounced hydrogen-out-of-plane band at ~957 cm$^{-1}$ (Fig. 1c).

Thus we confirmed that the yellow locust pigmentation is conferred by easily extractable β-carotene stabilized in the aqueous media by the protein. The pigmentation is preserved for a very long time, although the pigment loses the ability to be extracted into aqueous media with time.

### BBP reconstituted in β-carotene-producing *E. coli* cells is equivalent to the native yellow protein

Taking advantage of a collection of engineered *E. coli* strains synthesizing various carotenoid types, we then asked if the yellow protein can be reconstituted in *E. coli* cells synthesizing β-carotene[42] starting from a

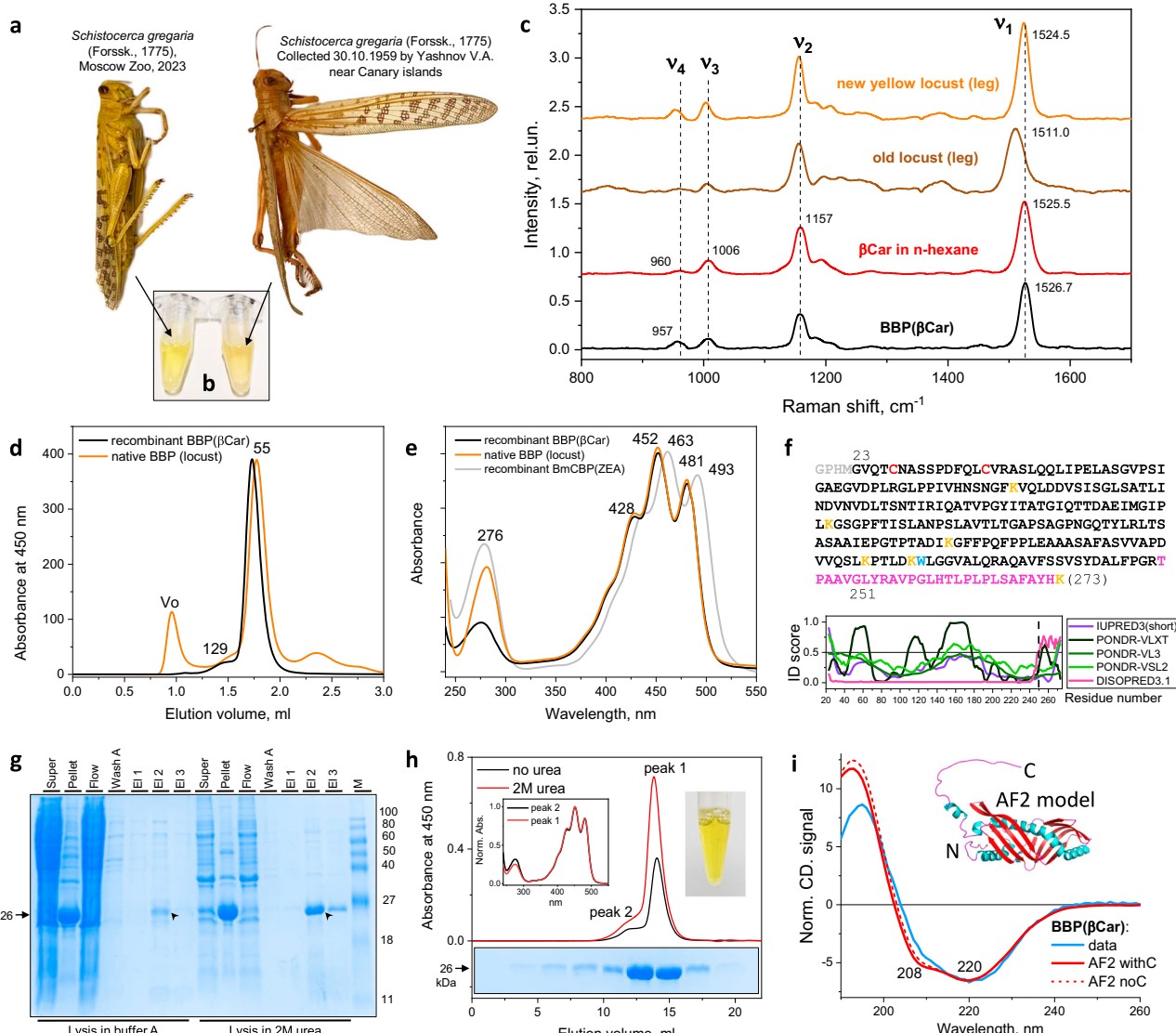

**Fig. 1 | BBP reconstituted in β-carotene synthesizing *E. coli* cells is equivalent to the native yellow protein extracted from locust cuticle. a** The photograph of two locust specimens used to extract yellow protein from the cuticles (panel **b**) and to collect Raman spectra (panel **c**). **b** The appearance of the centrifuged extracts into the SEC buffer obtained from the two locust specimens. **c** Raman spectra collected from the cuticles of the two locust specimens as compared with the spectra of β-carotene in n-hexane and in BBP. The main Raman bands are marked and the discussed maxima of the peaks are indicated (in cm$^{-1}$). **d** SEC profiles of the recombinant and native BBP (Superdex 200 Increase 5/150; 0.45 mL min$^{-1}$). Numbers indicate the apparent $M_w$ values in kDa obtained from column calibration. Vo - void volume. **e** Absorbance spectra of the recombinant and native BBP as compared with the spectrum of BmCBP complexed with ZEA. The main maxima positions are indicated in nm. **f** Amino acid sequence of the final BBP construct corresponding to the mature form devoid of the N-terminal signal peptide. Cys residues are red, Trp residue is light blue, Lys residues are orange, the C-terminal tail is pink. Below the sequence the prediction of the intrinsic disorder propensity is found. **g** SDS-PAGE of the fractions obtained upon BBP isolation after cell lysis in buffer A (20 mM Tris-HCl pH 8.0, 300 mM NaCl, 10 mM imidazole) without or then with 2 M urea. Super

and pellet are fractions obtained after sonication and centrifugation, of which the former was loaded onto HisTrap 5 mL column (Flow - flowthrough fraction), washed with buffer A (wash A) and then eluted by 500 mM imidazole (fractions El1, El2, El3 were collected consecutively). M designates protein markers with known $M_w$ (indicated in kDa on the right). Arrow on the left indicates the apparent $M_w$ of BBP with the His$_6$ tag. Barbed arrows indicate the effect of urea on BBP yield. **h** SEC profiles at 450 nm of BBP obtained by loading aliquots of BBP eluates from IMAC, pre-dialyzed against buffer A, for regular lysis and extraction (no urea) or extraction in the presence of 2 M urea (Superdex 200 Increase 10/300 column; 0.8 mL min$^{-1}$). The absorbance spectra for the peaks 1 and 2 registered throughout the elution profiles are shown in insert. Typical SDS-PAGE analysis of the fractions collected during the profiles is shown in the insert below. The appearance of the recombinant BBP sample loaded on the SEC column is shown in the insert. **i** Far-UV circular dichroism spectrum of the purified BBP as compared with the circular dichroism spectrum calculated by PDBMD2CD web server[47] for an Alphafold 2-derived model of BBP (colored by secondary structure element types) with or without the C-terminus.

synthetic BBP gene. The recently sequenced BBP gene contains an N-terminal signal sequence[28,37] and is predicted to be an extracellular protein with the score 0.898, lysosomal/vesicular with the score 0.285, and localized to endoplasmic reticulum with the score 0.093, according to Deeploc 2.0[43]. In fact, the natural cuticular BBP was identified already without the

N-terminal signal sequence[28,34]. Therefore, our protein design matched such mature BBP (residues 23–273) (Fig. 1f). Most of this construct is predicted as a folded protein, with an ambiguous status only for ~20–25 C-terminal residues (Fig. 1f). While IUPRED3[44] and PONDR[45] predictors suggested that this part of the protein is ordered, likely reflecting the high abundance of

hydrophobic (even several aromatic) residues, the DISOPRED3[46] prediction indicated some propensity to disorder.

The chosen construct showed very good expression levels in β-carotene-synthesizing *E. coli* cells. We observed much of the ~26 kDa protein in the yellow pellet and a relatively small amount in the soluble fraction (Fig. 1g). This fraction could be purified further and showed a major SEC peak almost coinciding with the native BBP extracted from the yellow locust cuticles (Fig. 1d). The visible absorbance spectra of the two protein versions were also identical (Fig. 1h). We hypothesized that BBP(βCar) could go to the insoluble fraction at least partly because of its interaction with components of cell debris and can be washed out. The addition of 2 M urea in the lysis buffer helped extract a major and much cleaner fraction of BBP(βCar) (Fig. 1g, h). Dialysis of BBP(βCar) against a urea-free buffer (or even against milliQ water) did not precipitate the target protein, which was purified further in the absence of urea. This sample was electrophoretically homogeneous, had ~2-fold higher yield and gave a SEC profile almost indistinguishable from that of BBP(βCar) extracted without urea (Fig. 1h). Both profiles also had a shoulder probably corresponding to a higher-order BBP oligomer. BBP was detected by sodium dodecyl-sulfate polyacrylamide gel electrophoresis (SDS-PAGE) in both peaks, and the absorbance spectra of the two were nearly identical (Fig. 1h, insert). The final BBP sample had a vibrant yellow color with a greenish hue due to the bound carotenoid (Fig. 1h, insert). Of note, mass-spectrometry analysis validated the amino acid sequence of BBP, yet no peptides corresponding to the hydrophobic C-terminus (Fig. 1f, pink) could be found (Supplementary Fig. 2). This suggests its complete in-cell cleavage, consistent with a lowered apparent $M_w$ (~26 kDa instead of 28.2 kDa expected for the His-tagged BBP). This implies that the BBP's C terminus is unlikely a structural element involved in folding of the central protein domain or in coordinating the carotenoid.

The far-UV circular dichroism spectrum of the final untagged BBP(βCar) displayed a minimum at 220 nm indicating well-folded protein (Fig. 1i). Notably, it slightly differed from a circular dichroism spectrum calculated by PDBMD2CD web server[47] on the basis of an unliganded BBP model predicted by Alphafold 2[48] (Fig. 1i), indicating the possibility of structural rearrangements within BBP upon carotenoid binding. Unfortunately, all our attempts to produce BBP apoform in *E. coli* failed, which allows us to speculate that BBP is unstable without the carotenoid.

## Recombinant BBP is a dimeric stress-tolerant protein

Given that the calculated $M_w$ for the BBP sequence is 26.2 kDa, similar apparent $M_w$ values of recombinant and native protein variants of ~55 kDa, (Fig. 1d) indicated that the major protein peak tentatively corresponds to a dimer. SEC-MALS independently determined absolute $M_w$ for BBP(βCar) as 49.4 kDa, i.e., twice as high as that for BBP monomer, simultaneously showing excellent sample monodispersity (Fig. 2a). Surprisingly, the SEC profile of BBP in various buffers remained unchanged, with the caveat that the addition of 2 M urea to the sample and to the running buffer slightly increased the apparent size of the protein. The addition of 0.05% deoxycholate (DOC) eliminated the shoulder tentatively corresponding to a BBP tetramer (Fig. 2b). BBP(βCar) showed a remarkable stability to pH variations, as incubation in 0.2 M acetic acid for 30 min and the following addition of excess of 10 M NaOH for another 30 min did not cause BBP precipitation nor changed its spectrochromatography profile (run in 2-(N-morpholino)ethanesulfonic acid (MES) pH 6.0). All tested factors did not affect the absorbance spectrum of BBP either (Fig. 2c). We occasionally observed a small spectral feature at 516 nm of intact BBP, which was absent in the presence of reducing agents. We tentatively suggest that it is linked with the intramolecular disulfide bond formation between the two N-terminal Cys residues and its influence on the chromophore, although this warrants further investigation.

We noticed that, while DOC removed the shoulder on the elution profile of BBP, it did not change the position of the main protein peak at ~1.75 mL (Fig. 2b). In order to test if other detergents would interfere with BBP dimerization, we subjected BBP mixtures with different concentrations of Tween 20, Triton X100, CHAPS and DOC to gel-electrophoresis under otherwise non-denaturing conditions (Fig. 2d). However, we could not detect any changes of the electrophoretic mobility of BBP caused by these detergents. In contrast, modified SDS-PAGE with optional sample heating and various SDS additions detached the carotenoid and produced the BBP band corresponding to a monomer (Supplementary Fig. 3). This supports the idea that BBP dimer is stabilized by the carotenoid. Interestingly, when subjected to chemical cross-linking with glutaraldehyde under conditions leading to efficient cross-linking of two control dimeric proteins, BBP migrated as monomer, cross-linked tetramer, and a barely detectable dimeric band (Fig. 2e). Low occupancy of the cross-linked dimeric species of BBP likely results from irregular distribution throughout the BBP molecule of lysine residues, which are modified by glutaraldehyde upon cross-linking (Fig. 1f).

We then set out to analyze the thermal stability of BBP. We used an approach which we termed SECmelt, especially suitable for colored proteins such as carotenoproteins, that is based on pre-incubation of protein samples at various temperatures for a fixed period of time, centrifugation and spectrochromatographic determination of the amount of native soluble protein (see "Methods" for details). In contrast to simple spectrophotometric determination of the protein concentration in the supernatant, SECmelt allows one to focus on a native protein peak found on the elution profile. SECmelt profiles for BBP at 280 nm or 450 nm absorbance consistently showed that this protein is rather thermostable, with the half-transition temperature ($T_{0.5}$) of ~68–70 °C (Fig. 2f). Incubation of BBP at 60 °C for 20 min decreased the amount of native protein by less than 10%. As proof-of-principle, we studied other carotenoproteins by SECmelt. A similar thermal stability ($T_{0.5} = 66$–67 °C) was found for the recently described Orange Carotenoid Protein 2 (OCP2) complexed with ECH, from *Gloeocapsa* sp. PCC 7418 inhabiting moderately hot springs[49] (Supplementary Fig. 4). Silkworm BmCBP exhibited biphasic SECmelt profiles with the half-transition temperatures slightly above 30 °C and of ~60 °C. This BmCBP sample contained an excess amount of the apoform along with the ZEA-bound protein, as seen from the Vis/UV absorbance ratio of less than 0.5 (Supplementary Fig. 4). According to differential scanning calorimetry, the apo-BmCBP has a much lower thermal stability (by more than 30 °C) than the ZEA-bound form, and, consequently, the apo-/holoprotein mixture has a bimodal thermogram with the shape depending on the ratio of the two forms[40]. This validated the original approach to assess the thermal stability of carotenoproteins, including the cases when the holoprotein is contaminated by the apoprotein, often having a lower thermostability[10,40]. Considering these results, it is safe to conclude that the reconstituted BBP samples are mostly free from the apoprotein, indirectly supporting the idea that the carotenoid is crucial for the stability of BBP and that the apoform of BBP is unstable.

## A peculiar local environment of β-carotene inside BBP

Spectral properties of carotenoids strictly depend on their molecular environment and the absorbance spectrum of a given carotenoprotein is dictated by the type of carotenoid bound. Reverse-phase HPLC confirmed that the recombinant BBP, like native BBP from locusts, indeed contains almost exclusively β-carotene (Fig. 3a and Supplementary Fig. 1). However, the absorbance spectrum of BBP(βCar) is very unusual for β-carotene in known environments. This prompted us to study spectral characteristics of BBP-carotenoid complexes in detail. We dissolved β-carotene in each of 13 different organic solvents and recorded the corresponding visible absorbance spectrum, to compare it with the spectrum of BBP(βCar) (Fig. 3b). As expected, the type of solvent influenced the position of the β-carotene spectrum greatly: from 447.5 nm in diethyl ether to 481 nm in carbon disulfide, constituting an almost 35 nm difference. The position of the β-carotene spectrum in BBP (452 nm) is hypsochromically shifted compared to that in most of the organic solvents studied and was the same as for example in acetone (Fig. 3b). This questions the universality of the assumption that upon protein binding the absorbance spectrum of the

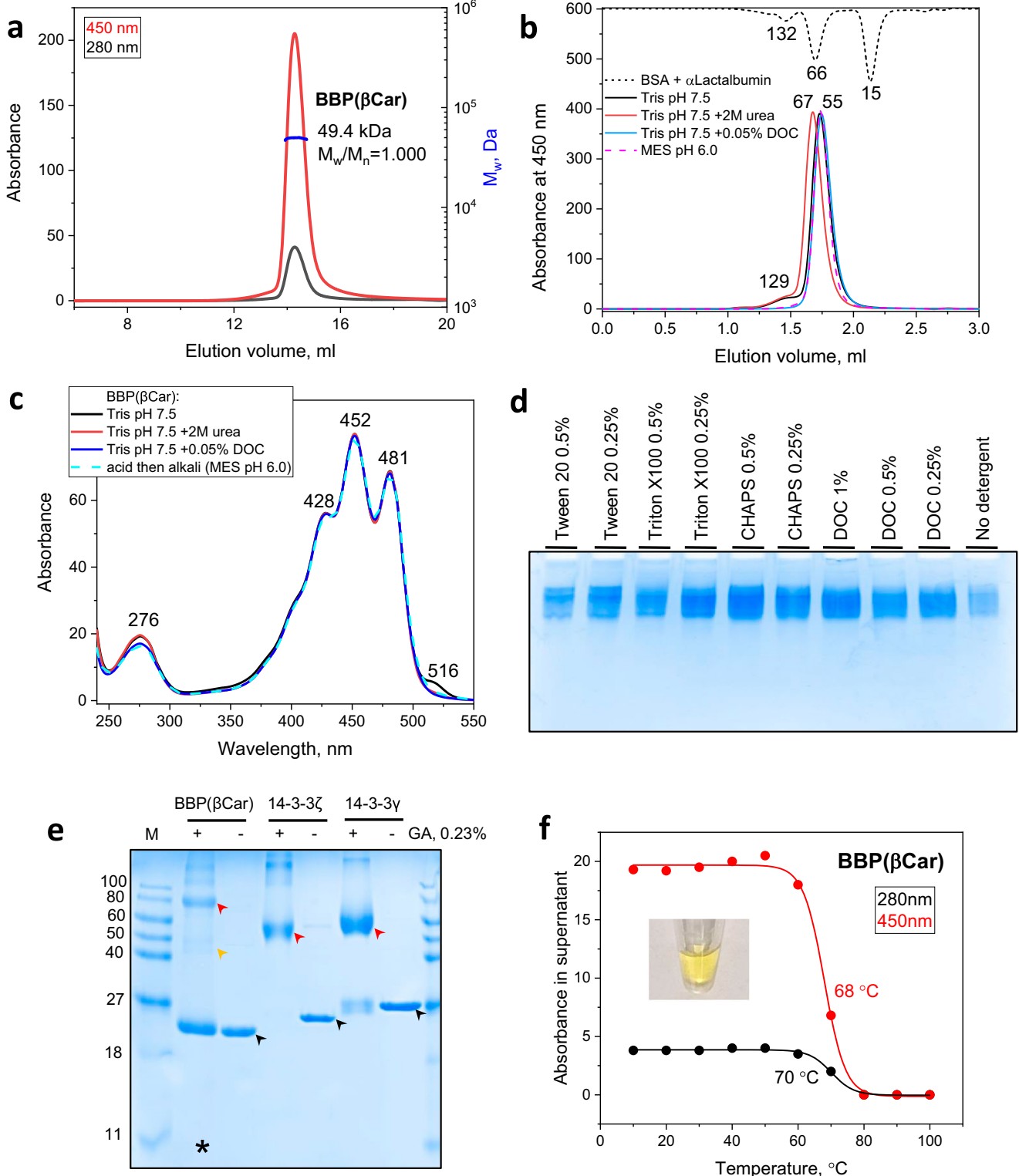

carotenoid is necessarily shifted bathochromically relative to that in organic solvents[9,21,50]. While the peak maxima of the β-carotene spectrum in BBP were very close to those for β-carotene in acetone, the vibronic structure of the former spectrum was much more pronounced (Fig. 3b). We also observed an induced circular dichroism of β-carotene upon binding to BBP. The shape of the UV/Vis circular dichroism spectrum was distinctly different from that of the BmCBP complex with ZEA, with the main extrema being nearly mirrored in the range of 250–375 nm by y = 0 axis (Fig. 3c).

This indicates a possible out of the plane torsion of the carotenoid polyene chain in the protein and/or the presence of cis-isomers. The induced UV/Vis circular dichroism of β-carotene in BBP provides an independent confirmation of unusual protein-pigment interactions. Indeed, β-carotene is an extremely hydrophobic hydrocarbon completely insoluble in water. But in complex with BBP, which most likely shelters β-carotene from the bulk aqueous environment, β-carotene can be concentrated up to 200 μM and higher, merely in a dilute buffer (Fig. 3d).

**Fig. 2 | Recombinant BBP is a dimeric stress-tolerant protein. a** SEC-MALS of BBP(βCar) (Superdex 200 Increase 10/300 column; 0.8 mL min⁻¹). **b** Analytical size-exclusion spectrochromatography on Superdex 200 Increase 5/150 column (0.45 mL min⁻¹) of BBP in 20 mM Tris pH 7.5 in the absence or in the presence of 2 M urea or 0.05% DOC in the sample and running buffer, or in 10 mM MES pH 6.0. In all cases, 150 mM NaCl was present in the sample and running buffer to minimize non-specific interactions. The elution profile of the standard proteins BSA and α-lactalbumin at 280 nm is flipped for the convenience of comparing protein peak positions. The apparent $M_w$ values (kDa) for BBP peaks (no tag) were calculated from column calibration. **c** Absorbance spectra of BBP in various chemical environments as recorded during spectrochromatography and represent spectra from the main chromatographic peak apex. Dashed cyan line represents the spectrum of BBP pre-conditioned first to acidic and then to alkalic conditions before the SEC run in MES pH 6.0. **d** Native PAGE analysis of the electrophoretic mobility of BBP in the absence or in the presence of various detergents added only in the corresponding

samples loaded on the gel. **e** Chemical cross-linking of BBP by glutaraldehyde analyzed by SDS-PAGE. Stable dimeric human 14-3-3 protein isoforms γ and ζ were used as positive controls. Black arrows indicate un-cross-linked monomeric species, red arrows indicate the main cross-linked forms. Orange arrow indicates the expected position of the cross-linked BBP dimer. Note that the cross-linked BBP sample was loaded on the lane (asterisk) in twice as large an amount as its un-cross-linked control. M designates protein standards with the known $M_w$ values (indicated in kDa). **f** Thermal stability of BBP assessed by SECmelt. Aliquots of BBP(βCar) were pre-incubated at different temperatures for 20 min, cooled, centrifuged, and their soluble fraction after heat treatment was studied by spectrochromatography. The amplitude of the chromatographic peak at the native protein position at either 280 nm or visible absorbance wavelength was used to plot the temperature dependences presented. The appearance of the sample used for the analysis is shown in insert.

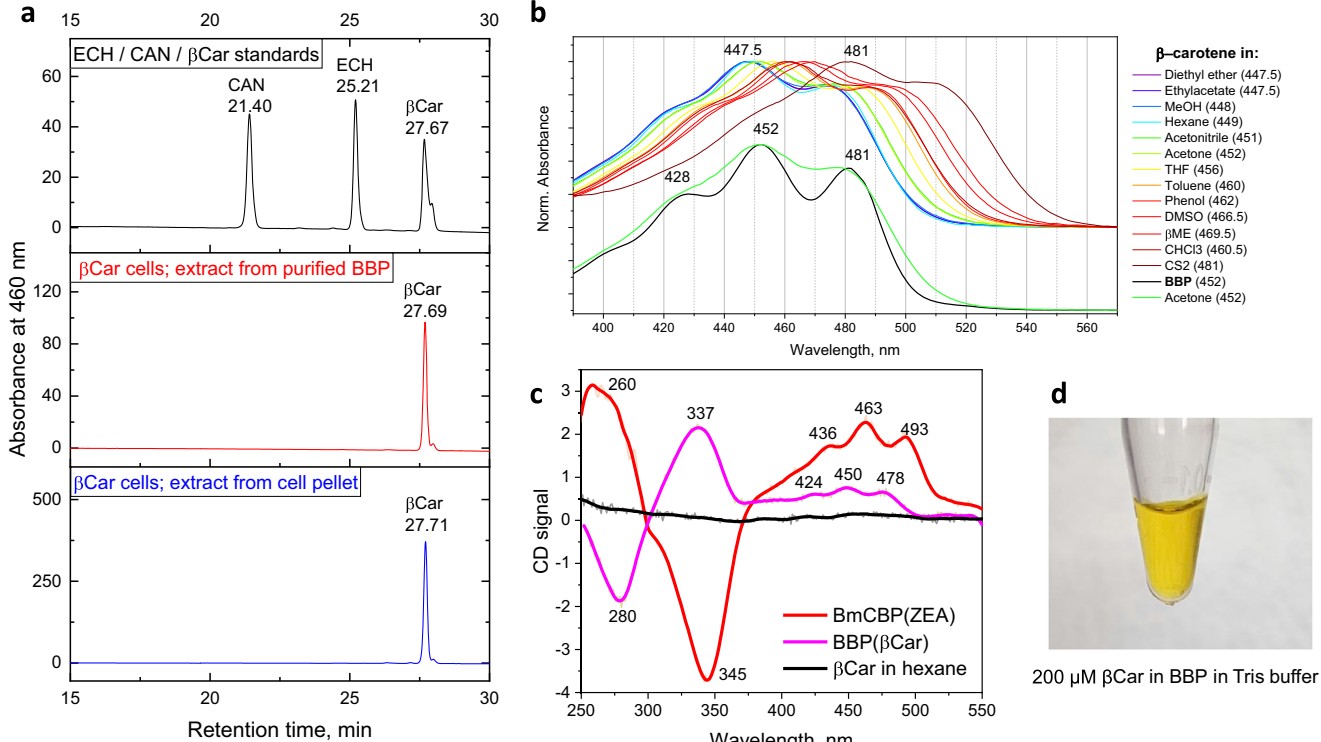

**Fig. 3 | Spectral signatures of β-carotene in BBP. a** HPLC analysis showing that β-carotene is the predominant carotenoid associated with BBP purified from β-carotene-synthesizing *E. coli* cells. **b** Absorbance spectra of HPLC-purified β-carotene in various organic solvents as compared with the absorbance spectrum of β-carotene in BBP. Positions of the main maximum are indicated in parentheses in nm.

Note the more pronounced vibronic structure of the BBP(βCar) spectrum. **c** UV/Vis circular dichroism spectra of β-carotene in n-hexane and in BBP. The UV/Vis circular dichroism spectrum of the BmCBP(ZEA) complex is shown for comparison. **d** The appearance of an aqueous 200 μM β-carotene solution in complex with BBP.

## BBP preferentially binds β-carotene, but can also bind xanthophylls

Our BBP samples characterized above contained exclusively β-carotene (Fig. 3a) and the ligand specificity of BBP remained undefined. To understand whether BBP can bind other carotenoids besides β-carotene, we expressed this protein in *E. coli* cells synthesizing a hydroxycarotenoid ZEA, or ketocarotenoids ECH and CAN (Fig. 4a–c). Given that all these xanthophylls are biosynthesized from β-carotene, we tested not only regular induction of protein expression at the stationary phase, but also late induction, which ensured that as much as possible xanthophylls are built up from β-carotene. Indeed, HPLC analysis indicated that early induction yields BBP containing almost exclusively β-carotene, and that a very low amount of ZEA had a chance to form (Fig. 4a). The absorbance spectra corresponding to the carotenoid peaks obtained via diode-array detection are shown in Fig. 4d, whereas their chemical structures are found in Fig. 4e.

Late BBP induction occurred at a point when the major carotenoid was ZEA and *E. coli* membranes contained no detectable amounts of β-carotene (Fig. 4b). Surprisingly, β-carotene was still the predominant carotenoid found in BBP isolated from these ZEA-producing cells, along with a tiny fraction of ZEA and some hydrophobic carotenoid with a retention time close to that of β-carotene. Based on its absorbance spectrum (Fig. 4d), we speculate that it may be γ-carotene, a precursor of β-carotene (Fig. 4e). Anyway, this experiment unequivocally indicated that BBP has a much higher preference for β-carotene over hydroxycarotenoid ZEA.

The situation with ketocarotenoids was different. Late induction of BBP expression permitted sufficient depletion of β-carotene and formation of a noticeable fraction of CAN and especially ECH, which resulted in the presence of all three carotenoids in the purified BBP sample (Fig. 4c). While nearly equal amounts of ECH and β-carotene were extracted by BBP, *E. coli* membranes contained much higher amounts of ECH than β-carotene. Even

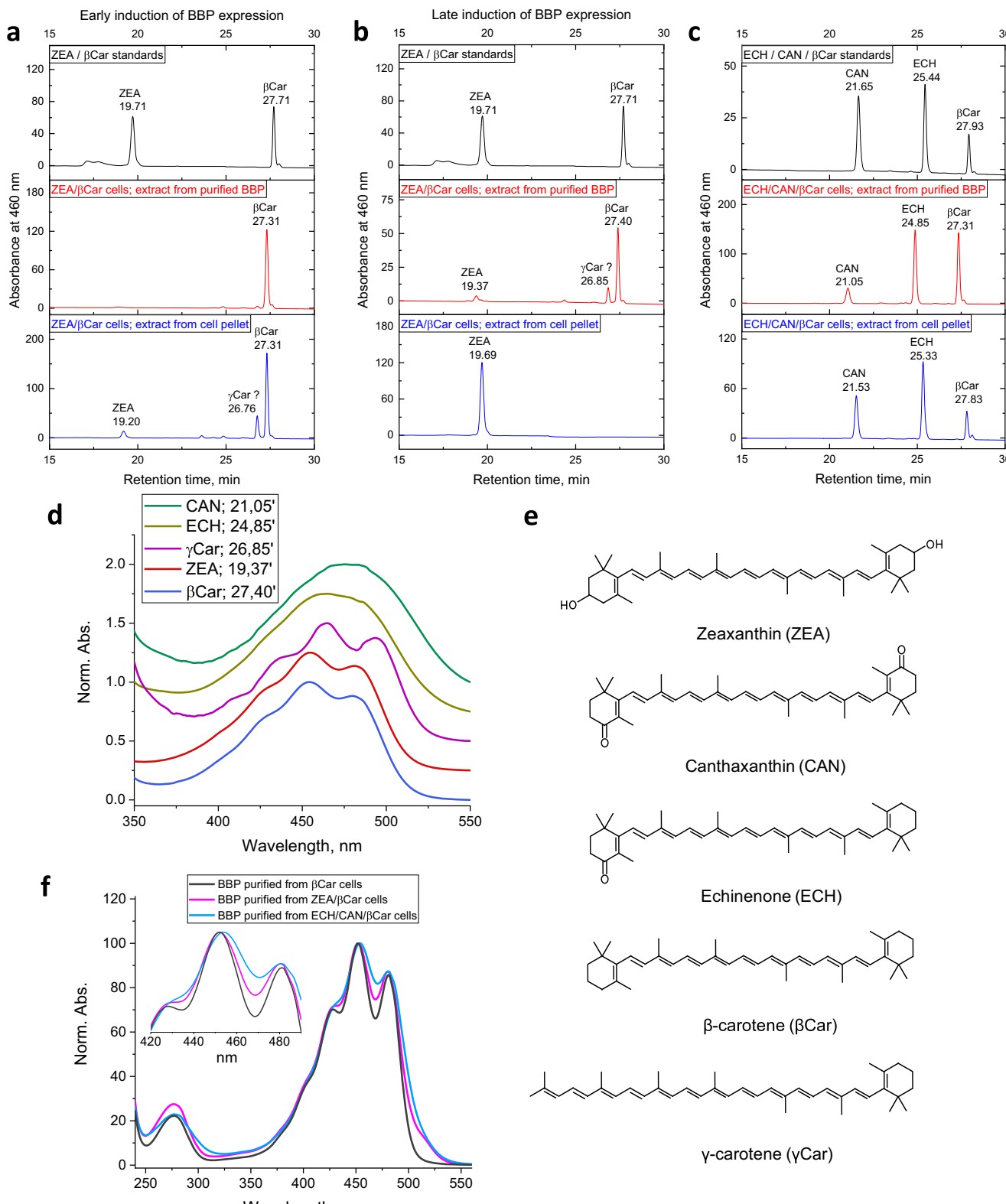

**Fig. 4 | BBP is capable of binding xanthophylls less selectively than β-carotene.**
**a–c** HPLC analysis showing the carotenoids extracted by BBP from the ZEA- (**a**, **b**) or ECH/CAN (**c**) synthesizing *E. coli* cells. The elution profiles of the carotenoid standards are on top, carotenoids associated with purified BBP are in the middle and carotenoids extracted from cell pellets after lysis are on the bottom. Early (**a**) or late (**b**, **c**) induction of BBP expression were studied. Note that β-carotene is the common precursor of other xanthophylls analyzed. The position of the carotenoid which we tentatively assign to γ-carotene is indicated by a question mark. **d** Absorbance spectra of HPLC-separated carotenoids obtained in the course of HPLC runs presented in panels (**a–c**). **e** Chemical structures of these carotenoids. **f** Absorbance spectra of the BBP samples purified from different carotenoid-producing *E. coli* cells as indicated.

more prominently, only a minor quantity of CAN could be found in BBP despite this doubly ketolated xanthophyll was more abundant in *E. coli* membranes than β-carotene (Fig. 4c). This unequivocally indicates that BBP avidly binds β-carotene but also exhibits capacity to bind ECH and, less efficiently, CAN. The absorbance spectra of BBP samples containing β-carotene with admixture of xanthophylls showed smoothening of the fine structure apparently due to the contribution from non-vibronic spectra of ECH and CAN (Fig. 4f).

### BBP reversibly binds to chitin and only very weakly interacts with chitosan

In the locust cuticle, BBP(βCar) complexes are within reach of the Raman microscope and easily extracted by SEC buffer (Fig. 1), which is in line with their extracellular localization[34,43]. Therefore, we hypothesized that BBP may interact with the chitin exoskeleton of the animals for their coloration. To test this, we exploited the marked differences in solubility of BBP and purified chitin flakes and incubated chitin in a 20 mM tris(hydroxymethyl) aminomethane (Tris) pH 7.5 buffer containing 150 mM NaCl (standard SEC buffer) alone or in the presence of BBP(βCar). In control samples, we mixed chitin with OCP2 carotenoprotein or with uncolored BSA (Fig. 5). Importantly, we did not use β-carotene as a control because it would in any case immediately precipitate in aqueous buffer and would therefore irreversibly stain the chitin pellet, not providing any relevant information.

In our experimental setup, we observed that in the case of carotenoproteins, the chitin pellet became colored (Fig. 5a, b). This clearly indicated migration of a proportion of proteins into a chitin fraction. Trying to assess the reversibility of such association, we washed the colored pellets by SEC buffer, then by 2 M urea and, finally, by SDS, and estimated the redistribution of proteins between the soluble and insoluble, chitin-associated fraction by SDS-PAGE (Fig. 5c). It turned out that only a minor portion of BSA moved from the soluble fraction to chitin and could be easily washed out by buffer. In contrast, both OCP2 and BBP, despite being unrelated carotenoproteins complexed with different carotenoids, partly disappeared from the soluble fraction (somewhat more efficiently in the case of BBP), indicating their attachment to chitin. Given this similar behavior of the two carotenoproteins, we suppose that the BBP-chitin complexes are rather non-specific and dissociable, which may be relevant for the rapid and reversible color change of the locust cuticles.

Since BBP interacted with chitin, we hypothesized that it could also interact with a soluble polysaccharide chitosan differing from chitin by the absence of N-acetyl groups (Fig. 5d). This hypothesis was tested by SEC with absorbance spectrum and refractive index detection. Increasing amounts of chitosan demonstrated gradual rise of the refractive index throughout the whole working range of the elution profile, suggesting reasonable amounts for the assay (100 μg). In the assay, individual BBP eluted as a single symmetrical peak with the typical visible and UV absorbance (Supplementary Fig. 5). Comparison of the elution profiles for the BBP-chitosan mixtures followed by multiparameteric detection with those of individual chitosan, or BBP, or their algebraic sum, has allowed us to detect only minor changes. This indicated a very unstable BBP-chitosan association, unlike that with chitin (Fig. 5).

## Discussion

While β-carotene binding to BBP has been proposed and broadly documented[27,29,36], direct stringently controlled studies on BBP(βCar) complexes have not been undertaken. In the present study, we successfully reconstituted and comprehensively characterized BBP, a rare example of soluble β-carotene-binding proteins. Crucially, the purified recombinant BBP complexed with β-carotene had the absorbance spectrum and SEC elution profile equivalent to those displayed by the native protein isolated from yellow locust cuticles (Fig. 1). We have demonstrated that 2 M urea is sufficient to extract much of the folded and functional BBP from the *E. coli* cell debris, preserving the absorbance spectrum with the high Vis/UV absorbance ratio of ~4 characteristic of the same protein extracted in the absence of urea. While urea is often used as a denaturing agent, this result is

not unexpected since previous works demonstrated that tested proteins preserve native-like conformation in 2 M urea[51–53]. The recombinant dimeric BBP is saturated by carotenoid and is free from the apoprotein, probably because the latter is rather unstable. The far-UV circular dichroism spectrum of BBP(βCar) is typical of a folded protein, yet there are noticeable differences compared with the spectrum calculated from the structural model of unliganded BBP. This may be indicative of structural rearrangements of BBP upon carotenoid binding. Since carotenoid substantially stabilizes the dimeric protein (Fig. 2), one can expect an expanded dimerization interface and even the formation of a domain-swapped architecture, which would be in line with the extreme stability of BBP(βCar). While structural biology methods could resolve this enigma, our attempts to obtain diffraction-quality crystals for this complex have not been successful so far.

Interestingly, we could not confirm that BBP is an efficient binder of hydroxycarotenoids, as was claimed by Sugahara et al.[37], and in this respect BBP is distinctly different from carotenoproteins BmCBP[21,22,40] and AstaP[9,10]. We demonstrated that BBP has the fundamental ability to bind ketocarotenoids ECH and CAN, although much less selectively than β-carotene (Fig. 4). This may be physiologically relevant for the locusts feeding on plant petals rich in different carotenoid types. Nevertheless, we show that BBP is a genuine β-carotene-binding protein, solubilizing β-carotene as efficiently as BmCBP solubilizes ZEA, up to at least several hundred micromolar carotenoid concentration in aqueous medium (ref. 22 and Fig. 3d). This creates a nice opportunity for the future applications of BBP.

Noteworthy, comparison of the Raman spectra (Fig. 1) from BBP(βCar) and from the yellow locust cuticles in situ revealed the remarkable similarity between the two, and differences compared with the Raman spectrum of free β-carotene. This confirmed that the cuticular β-carotene is protein-bound. Given the high Vis/UV absorbance ratio of BBP(βCar) and instability of the protein apoform, we propose that BBP is not functioning as a shuttle for β-carotene but is rather a terminal depot for β-carotene, and that together these two entities form the water-soluble macromolecular pigment. This resembles β-crustacyanin of lobster shells[11] or the blue carotenoprotein from sea sponge[14] also representing macromolecular carotenoprotein pigments. In contrast, the coloration of silkworm cocoons is mediated by the transport function of BmCBP[20] transferring carotenoids that are deposited in the silk.

Since BBP(βCar) can be easily extracted from the yellow cuticles by buffer in the absence of denaturants (Fig. 1), we hypothesized that BBP has an affinity to chitin. Although chitin-binding sites were predicted on BBP[28], such hypothesis has not been tested. We have shown that BBP tightly but reversibly interacts with chitin (Fig. 5), whereas the interaction with chitosan is negligible (Supplementary Fig. 5). This may indicate that the chitin's N-acetyl groups are important for the association (Fig. 5). Of note, BBP sequence has a higher-than-average composition of short-chain amino acid residues, many of which (Ser, Thr, and Ala) decorate the surface of the structural model of BBP, which can be beneficial for the BBP attachment to chitin.

The interaction with chitin is not specific to BBP as the other carotenoprotein, OCP2 (but not BSA used as a control), was capable of chitin binding under near-physiological conditions (pH 7.5, 150 mM NaCl). Importantly, chitin binding of BBP and OCP2 was not mediated by other proteins as none were detected in our chitin preparation by extraction and SDS-PAGE (Fig. 5). The interaction of BBP with chitin points to an interesting potential explanation of our observation that BBP expressed in *E. coli* can be efficiently extracted by 2 M urea from co-sediments with the cell debris (Fig. 1) because bacterial cell wall peptidoglycan murein contains N-acetylglucosamine moieties[54] similar to those found in chitin (Fig. 5).

BBP has been described as a non-typical cuticular protein[34]. Modern approaches based on deep-learning algorithms unequivocally predict extracellular localization of BBP[43,55]. In line with its secretion-related coloration function, N-terminal sequencing of native BBP(βCar) revealed the lack of the N-terminal signal peptide[34] (Fig. 5). Immunogold staining detected BBP(βCar) within the pigment granules of epidermis and in the extracellular space in the integument of gregarious locusts[29,36]. Therefore,

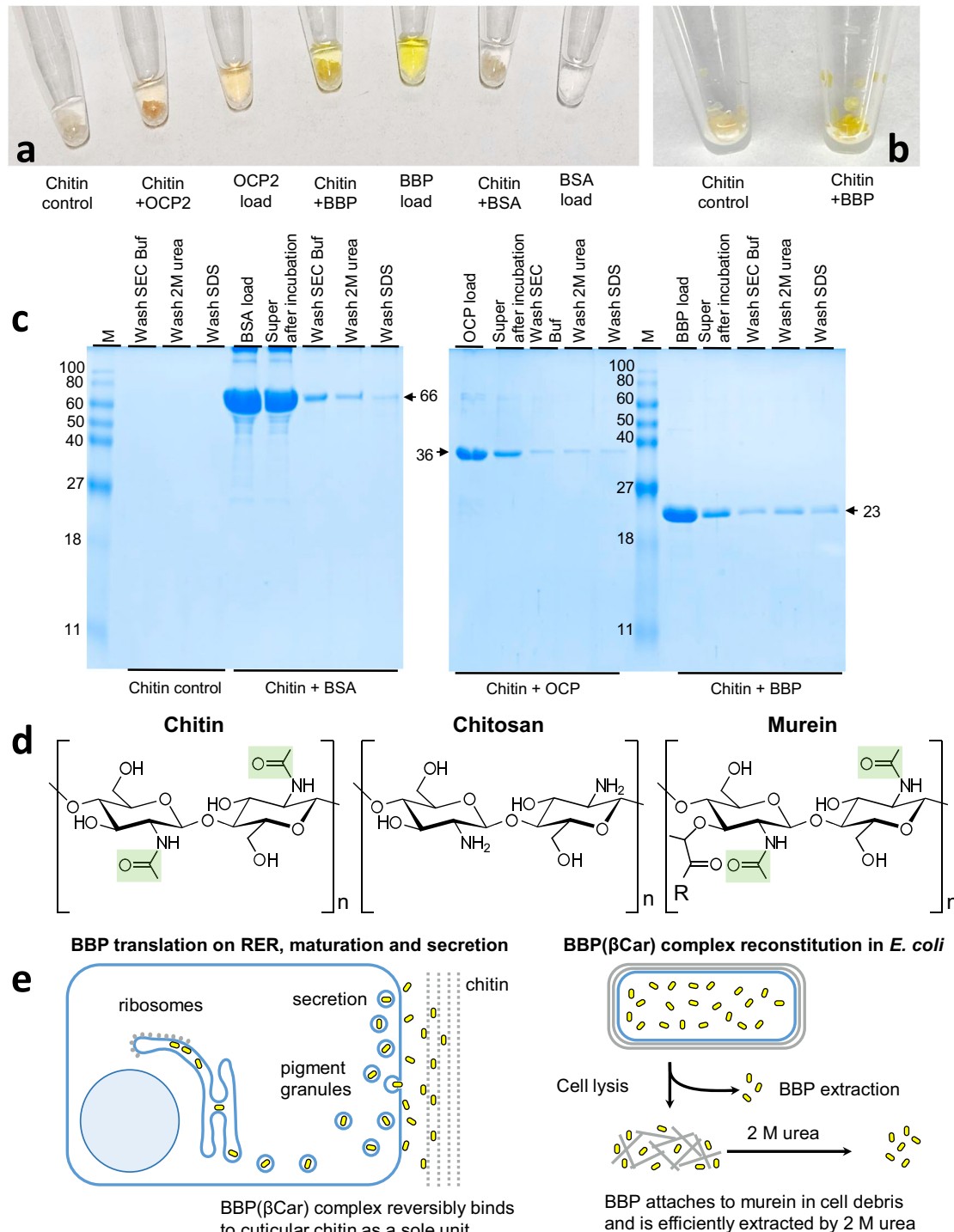

**Fig. 5 | BBP binds to chitin. a** The appearance of samples at the end of the incubation of BBP(βCar), OCP2(ECH), or BSA in the absence or in the presence of chitin. Chitin in the SEC buffer instead of these proteins is shown as a control. **b** The insoluble chitin fraction incubated in the presence of BBP is intensely colored yellow. The sedimented fractions are shown. **c** SDS-PAGE analysis of the results of the chitin-binding assay. Pre-incubated chitin-protein complexes were sedimented and then washed with SEC buffer, 2 M urea and SDS, consecutively. Control lanes show that chitin on its own contains a negligible amount of proteins. M designates protein standards with the known $M_w$ values (indicated in kDa). The apparent $M_w$ of BBP

(no tag), OCP2 and BSA are shown by arrows. **d** Structures of chitin, chitosan and murein fragments. The N-acetyl groups tentatively used for BBP attachment are highlighted by green. R - tetrapeptide of murein. **e** Schematic showing the presumed mechanism of BBP(βCar) complexes accumulation in pigment granules and cuticular chitin coloration by secretion of BBP(βCar) representing the yellow pigment (left). RER - rough endoplasmic reticulum. Schematic showing BBP(βCar) reconstitution in *E. coli*, non-specific attachment to the murein component of cell debris after lysis, and extraction by 2 M urea, which disrupts BBP(βCar)-murein interactions (right).

secreted BBP(βCar) is likely the terminal functional stage of this protein, as secretion predisposes BBP for attachment to cuticular chitin (Fig. 5e). Nevertheless, part of the yellow coloration is apparently contributed by BBP(βCar) residing in the pigment granules within the epidermal cells[29,36]. Large amounts of BBP(βCar) pigments are obviously required for the locust coloration. As recently discovered, those are overexpressed in response to external gregarization-associated stimuli activating PKCα, which phosphorylates the ATF2 transcription factor to induce ATF2 binding to the BBP promoter and BBP transcription[36]. The carotenoids are likely supplied for complexation with BBP by mobilizing the storage form in a fat body and other organs[27,34].

The remarkable heat and chemical stability of BBP in complex with the abundantly available β-carotene can be beneficial for the pigment preservation under the harsh environment conditions potentially experienced by the desert locusts, including desert heat and sunlight exposure, which would be damaging for the unbound β-carotene in the absence of BBP. Such durability of the characterized carotenoprotein complex is in line with the detection of signature carotenoid lines in the Raman spectrum recorded from the 64-year-old locust male from the insect collection (Fig. 1). In the future, it would be interesting to see if the described coloration mechanism is used by other insect species, especially given that many of them feed on plants and deal with carotenoids in abundance.

## Methods
### Materials
Organic solvents were of the best available purity and quality and were purchased from Khimmed company (Russia). All aqueous solutions in the study were prepared on the milliQ water (18.2 MΩ/cm). Bovine serum albumin (BSA) was from Serva (99%). Glutaraldehyde (25%) was from Sigma.

Chitin and low molecular weight chitosan were kindly provided by Prof. Valery P. Varlamov (Institute of Bioengineering, FRC of Biotechnology of the Russian Academy of Sciences). Animals were kindly provided by Dr. O.S. Korsunovskaya (Entomology Department, Faculty of Biology, Lomonosov MSU). A naturally died *Schistocerca gregaria* (Forssk., 1775) male was grown ex-culturae in the Moscow Zoo. An old specimen of *Schistocerca gregaria* (Forssk., 1775) male was taken from the collection of the Entomology Department, Faculty of Biology, Lomonosov MSU. According to the records, the animal was collected on 30.10.1959 by V.A. Yashnov near the Canary islands on the research vessel "Lomonosov". Euthanized with HCN, the specimen was dried and preserved in darkness in the presence of naphthalene and vaportrine.

### Prediction of the intrinsic disorder propensity of BBP
The sequence of the mature form of BBP (residues 23–273) was used as input to predict disordered regions using PONDR[45], DISOPRED3[46], and IUPRED3[44] and default parameters.

### Cloning, protein expression, and purification
The cDNA corresponding to residues 23–273 of the putative β-carotene binding protein from *Schistocerca gregaria* (termed BBP; NCBI ID: XP_049831059.1, which is identical to Uniprot A0A6N3ISN1) was codon-optimized for expression in *E. coli*, synthesized by Kloning Fasiliti (Moscow, Russia) and cloned into the modified pET28-His-3C vector (kanamycin resistance) using the *NdeI* and *XhoI* restriction sites. The pET28-His-3C plasmid contains an N-terminal His6 tag cleavable by human rhinovirus 3 C protease leaving extra N-terminal GSHM… residues after cleavage.

The ZEA-bound form of the Carotenoid-Binding Protein from *Bombyx mori* (termed BmCBP; residues 68–297 of Uniprot Q8MYA9) and the ECH-bound form of the Orange Carotenoid Protein 2 from *Gloeocapsa* sp. PCC 7428 (OCP2; residues 1–318 of Uniprot K9XH36) were obtained as electrophoretically homogeneous preparations in previous works[40,49]. In the final form, all proteins had His-tags cleaved off. All constructs were verified by DNA sequencing in Evrogen (Moscow, Russia) (BBP sequencing results are found in Supplementary Fig. 2).

The BBP holoprotein was obtained via expression in different carotenoid-synthesizing *E. coli* strains. For constitutive β-carotene (βCar) synthesis, *E.coli* cells were transformed with the pACCAR16ΔcrtX plasmid (chloramphenicol resistance) containing the cluster of *crtY*, *crtI*, *crtB*, and the *crtE* genes from *Pantoea ananatis* (formerly known as *Erwinia uredovora*)[42]. For constitutive ZEA production, the pACCAR25ΔcrtX plasmid (chloramphenicol resistance) containing the cluster of *crtY*, *crtI*, *crtB*, *crtZ*, and the *crtE* genes from *Pantoea ananatis* was used[42]. The product of the *crtE* gene, which is geranylgeranyl-pyrophosphate synthase, forms geranylgeranyl-pyrophosphate from isopentenyl-pyrophosphate and farnesyl-pyrophosphate, natural metabolites of *E. coli*. The product of the *crtB* gene, which is phytoene synthase, uses two geranylgeranyl-pyrophosphate blocks to form phytoene, which is then converted to lycopene by the *crtI* gene product, phytoene desaturase. The *crtY* gene product, which is lycopene cyclase, forms βCar from lycopene. Finally, the product of the *crtZ* gene, βCar hydroxylase, converts βCar in two hydroxylation steps at the 3 and 3' position of the β rings, to ZEA[42].

Ketocarotenoid production was achieved using *E. coli* cells carrying the pACCAR25ΔcrtXZcrtO plasmid (chloramphenicol resistance). This plasmid harbors the cluster of the *crtY*, *crtI*, *crtB*, *crtE,* and *crtO* genes from *Pantoea ananatis*, which enables ECH expression[42,56]. Alongside ECH, CAN is also produced in this strain, which may contain various amounts of ECH, CAN, and βCar depending on expression conditions[57,58].

Protein expression was done in C41(DE3) *E. coli* cells transformed by BBP-coding pET28-His-3C vector and the corresponding plasmids for carotenoid synthesis. It was noted that different amounts of oxidized carotenoids formed depending on the cultivation time. For the differential carotenoid synthesis, two separate protein induction schemes were performed for the same strain. In an early induction scheme, cells were grown to $OD_{600} = 0.6$ and then isopropyl β-D-thiogalactopyranoside was added up to a final concentration of 0.25 mM and protein expression lasted for 24 h at 28 °C. In a late induction scheme, cells were grown overnight at 37 °C with robust aeration, then the culture was chilled and freshly prepared media ($\sim^1/_3$ by volume) was added. Only after this, isopropyl β-D-thiogalactopyranoside was added to a final concentration of 0.25 mM and protein expression lasted for 12 h at 28 °C. Our attempts to express BBP apoform using the same plasmid that was used to produce the holoform resulted in only subtle expression yield, or the apoprotein was simply degraded upon the expression.

Recombinant BBP holoprotein was first extracted using regular lysis and then was repeated after the addition of 2 M urea into the lysis buffer. Each extract was subjected to exhaustive centrifugation and the yellow supernatants were chromatographically purified using the subtractive immobilized chromatography (IMAC) with the His6 tag removal stage in between. For the second extract, we added 2 M urea to all buffers during the first IMAC step, whereas urea was removed by dialysis prior to 3 C proteolysis and was not added afterward. The yellow protein remained completely soluble once extracted from the cell debris by 2 M urea. The last purification step consisted in semi-preparative gel-filtration on a Superdex 200 Increase 10/300 column (Cytiva), equilibrated by 20 mM Tris-HCl pH 7.5 buffer containing 150 mM NaCl and 3 mM βME, run at a 0.8 mL min⁻¹ flow rate. Protein elution was followed by a Prostar 335 diode-array detector (Varian). Run files were converted into wavelength-time matrices using the in-house developed python script. Electrophoretically homogeneous BBP fractions with consistent Vis/UV absorbance ratios of more than 4 were pooled and used for further studies. Purified proteins were stored at −80 °C. The final untagged BBP preparation giving an apparent $M_w$ of 23 kDa on SDS-PAGE was excised from the gel and subjected to in-gel trypsinolysis and mass-spectrometry identification on a Bruker UltraflexTreme mass-spectrometer, which revealed peptides covering all BBP sequence but the C-terminus (Supplementary Fig. 2).

We additionally purified native BBP from the yellow male locust presented in Fig. 1a, approximately 4 days after its natural death (until which it was stored at 4 °C). Purification from the native source included dissection and grinding in a mortar of part of the yellow integument and mere

extraction with 20 mM Tris-HCl buffer pH 7.5 containing 150 mM NaCl, 3 mM βME and 0.5 mM PMSF upon 1.5 min of sonication on ice. The extract was clarified by centrifugation for 20 min at $21,400 \times g$ and 4 °C, filtered through 0.22 μm filter and then analyzed (90 μL) by spectro-chromatography on a Superdex 200 Increase 5/150 column (Cytiva) at 0.45 mL min$^{-1}$. We also tried to extract into SEC buffer carotenoprotein(s) from the 64-year-old locust from the collection, but this was not successful probably due to more tight immobilization of BBP in the integument, requiring more harsh treatments which we did not attempt.

### Reverse phase high-performance liquid chromatography (HPLC)
Carotenoids produced by *E. coli* cells or extracted from protein were analyzed with the help of a Nucleosil C18 4.6*250 column run at 1 mL min$^{-1}$ and column oven temperature set to 28 °C. For extraction, 100 μL of a protein sample or an aliquot of cell precipitate were mixed with 100 μL acetone and 100 μL hexane, then vortexed and centrifuged until phase separation was observed. The carotenoid-containing upper hexane fraction was dried under the gaseous nitrogen stream, redissolved in 25 μL acetone, and injected onto the C18 column. Carotenoid standards (ZEA, CAN, ECH, and βCar) were injected separately as premixes. The elution was followed by absorbance at 460 nm and the following gradient structure: 0–5 min 70% acetone in water, 5–25 min gradient of 70->100% acetone in water, 25–30 min 100% acetone, 30–32 min 100->70% acetone, then 70% acetone until 40 min. The HPLC runs were carried out on a Prostar 210 system (Varian) equipped with a Prostar 335 diode-array detector enabling to record the 240–900 nm absorbance spectrum of the carotenoids eluted.

### β-carotene purification and visible spectrum registration
βCar was obtained from βCar-producing *E. coli* cells transformed with the pACCAR16ΔcrtX plasmid (chloramphenicol resistance) with constitutive expression of βCar synthesis pathway genes. Cells were cultured overnight in LB media on 30 °C and then centrifuged. Carotenoids were extracted from cell paste using acetone and concentrated on a rotary evaporator under vacuum and 55 °C. After acetone evaporation terminated and remaining water removed, the mixture of cell lipids and carotenoids was dried under gaseous nitrogen steam and redissolved in a small aliquot of hexane.

To get rid of cell lipids, chromatography on silicagel was performed. A glass chromatography column with a sintered glass filter was filled with silicagel 60 (0.04–0.063 mm, Macherey-Nagel, Germany) and equilibrated with hexane. Concentrated carotenoid solution in hexane was loaded onto the column and washed with 1 column volume of hexane. The βCar fraction was eluted with hexane-acetone mixture (95:5 by volume). An enriched βCar fraction was dried under gaseous nitrogen steam and redissolved in a small aliquot of acetone, centrifuged and injected onto a C18 column as described above. The main peak with the retention time and absorbance spectrum characteristic of the βCar standard was collected and assumed to contain pure βCar.

Aliquots of purified βCar were dried under gaseous nitrogen steam and dissolved in the desired organic solvent for visible absorption spectra measurements in a 10 mm quartz cuvette on a NanoPhotometer NP80 (Implen, Germany).

### Spectrochromatography
Spectrochromatography enables simultaneous analysis of the hydrodynamic radius/size of the protein and its spectral characteristics and is especially useful in the case of carotenoproteins[9,22,40]. BBP was first pre-conditioned to various buffers (at least 20 min at room temperature), clarified by centrifugation for 5 min at $21,400 \times g$ at 4 °C and then loaded onto a Superdex 200 Increase 5/150 column (Cytiva) pre-equilibrated with the appropriate buffer. We performed runs in the standard SEC buffer (20 mM Tris-HCl, pH 7.5, containing 150 mM NaCl and 3 mM βME) or in the SEC buffer supplemented with either 2 M urea or 0.05% DOC. Alternatively, a more acidic buffer, 10 mM MES, pH 6.0, containing 150 mM NaCl was used. Also, we subjected to spectrochromatography aliquots of the clarified soluble extracts obtained from the locust cuticles.

The samples (90 μL) stored at 6 °C were injected using Vialsampler (G7129A, Agilent). The flow rate was 0.45 mL min$^{-1}$. The runs were performed on an Agilent 1260 Infinity II chromatography system equipped with a detector system including a WR diode-array detector (G7115A, Agilent) and a refractometric detector (G7162A, Agilent, operated at 35 °C). The elution profiles were followed by recording full absorbance spectrum in the 240–750 nm range with 1-nm precision (4 nm slit width) and a 2.5 Hz frequency. The spectral accuracy was verified by the use of the built-in Holmium oxide filter in the G7115A absorbance detector enabling wavelength calibration/correction. Each experiment was performed at least three times, and the most typical results are presented.

To determine apparent $M_w$ of the protein peaks observed, the column was pre-calibrated using α-lactalbumin (15 kDa), bovine serum albumin monomer (66 kDa), dimer (132 kDa), and trimer (198 kDa).

### SEC-MALS
Absolute $M_w$ of BBP(βCar) was determined independently of a protein shape and possible non-specific interactions with the chromatographic resin by SEC coupled with multi-angle light scattering (SEC-MALS). BBP(βCar) (90 μL) was loaded on a Superdex 200 Increase 10/300 column (Cytiva) pre-equilibrated with 20 mM Tris pH 7.5, 150 mM NaCl, 3 mM βME. The flow rate was 0.8 mL min$^{-1}$. The run was carried out on an Agilent 1260 Infinity II chromatography system equipped with a WR diode-array detector (G7115A, Agilent), a miniDAWN detector (Wyatt) and a refractometric detector (G7162A, Agilent, operated at 35 °C), connected one-by-one in this order. The post-column 0.1 μm filter was used to reduce the noise in the miniDAWN detector. The elution profiles were followed by absorbance at 280 and 450 nm as well as dRI. The data processing was performed in Astra 8.0 software (Wyatt) using the refractometric detector as a concentration source (dn/dc was taken equal to 0.186). Normalization of the static light scattering signals at different angles detected by miniDAWN and alignment of the light scattering and refractometric signals, to account for the connection capillary between the detectors, were done during a pre-run of a BSA standard (Wyatt), for which $M_w$ was determined as 66.1 kDa ($M_w/M_n = 1.000$) at the same values of calibration constants and other parameters as were then used for BBP(βCar). The monodispersity ($M_w/M_n$) was determined upon $M_w$ averaging across the main protein peaks.

### Circular dichroism spectroscopy
BBP(βCar) or BmCBP(ZEA) holoproteins were equalized by absorption in the visible range of the spectrum ($A_{460} = 2.5$, i.e. BBP 0.5 mg mL$^{-1}$, and BmCBP 5 mg mL$^{-1}$) and dialyzed overnight at 4 °C against 20 mM phosphate buffer, pH 7.2, and centrifuged for 10 min at 4 °C and $21,400 \times g$ before measurements. The far-UV circular dichroism spectrum of BBP was recorded at 20 °C in the range of 180–280 nm at a 20 nm min$^{-1}$ rate with 0.5 nm steps in 0.1 mm quartz cuvette on a Chirascan circular dichroism spectrometer (Applied Photophysics) equipped with a temperature controller. Then the signal from the buffer was subtracted. The experimentally obtained far-UV circular dichroism spectrum of BBP(βCar) was compared with the spectrum calculated from the structural model of BBP predicted by Alphafold 2[48]. Circular dichroism spectrum calculation was done using PDBMD2CD[47].

Circular dichroism spectra in the UV and visible region were obtained for the 250–550 nm range, in a 0.5 mm quartz cuvette, at a 20 nm min$^{-1}$ rate with 0.5 nm increments. βCar solution in n-hexane with the same absorption in the visible range of the spectrum ($A_{460} = 2.5$) was used as a reference.

### Raman spectroscopy
We used Raman spectroscopy to compare the spectral signatures of β-carotene in BBP, in hexane and in the cuticle of the two specimens of desert locusts kindly provided by Dr. Olga S. Korsunovskaya (Entomology Department, Faculty of Biology, Lomonosov MSU). BBP and β-carotene in hexane were studied in quartz capillaries in a liquid form, whereas the legs of the locusts were measured in a solid form. The measurements were done on a Ntegra Spectra confocal

microscope (NTMDT, Russia) equipped with a CCD spectrometer. Linearly polarized 532-nm laser light was focused on the object using 20X (numerical aperture = 0.4) or 50X (numerical aperture = 0.8) Olympus (Japan) lenses. The average laser power at the lens exit was about 0.5 mW. The accumulation time of single spectra varied from 10 s to 2 min, depending on the type of measurements and the strength of the signal.

## Analysis of the oligomeric state of BBP by gel electrophoresis and chemical cross-linking

BBP(βCar) exhibits the apparent $M_w$ of 55 kDa upon SEC, which is close to protein dimer ($M_w$ calculated from sequence is 26.2 kDa). To have an independent evidence toward the oligomeric state of BBP (calculated pI ~6.1), we pre-incubated protein samples in the absence or the presence of Tween 20 (0.25, 0.5%), Triton X100 (0.25, 0.5%), CHAPS (0.25, 0.5%) or DOC (0.25, 0.5, 1%) and subjected them to native PAGE in the homogeneous glycine-Tris system at pH 8.6[59]. In addition, we analyzed BBP mobility by SDS-PAGE under modified conditions. One sample was processed regularly being heated in the presence of SDS and others were not heated and were premixed with the same or 5 times lower SDS concentration. In control, BBP was subjected to electrophoretic separation without SDS and with no heat treatment, to preserve any oligomeric species present. The gel contained regular SDS concentration.

Cross-linking with glutaraldehyde (final concentration 0.23%, 15 min at 30 °C) was carried out in 20 mM 2-[4-(2-hydroxyethyl)piperazin-1-yl] ethanesulfonic acid (HEPES) buffer pH 7.5 containing 150 mM NaCl, to independently examine the probable self-association of BBP. As a positive control we used the same conditions to cross-link the stably dimeric human 14-3-3 protein obtained in previous work[60]. Separation of the cross-linked proteins and their uncross-linked controls was carried out by SDS-PAGE[61].

## Thermostability measurements using SECmelt technique

To assess thermal stability of BBP and other carotenoproteins, we adapted the method based on a combination of dosed sample heating and fluorescence-detection size-exclusion chromatography of the unprecipitated proteins remaining in the supernatants[62]. In our case, fluorescence detection, however, was not needed because carotenoproteins exhibit strong absorbance in the visible spectral region. Due to this peculiarity, the modified method, which we called SECmelt, requires low amounts of sample and can be successfully applied to other similar proteins. In our setup, BBP complexed with βCar, BmCBP complexed with ZEA but containing an excess amount of apoprotein, and GcapOCP2 complexed with ECH were first diluted by SEC buffer (20 mM Tris-HCl pH 7.5, 150 mM NaCl, 3 mM βME) to concentration when the sample color was still discernible by eye and then split into 20 µL aliquots. Each aliquot was incubated at a specific temperature from 10 to 90 °C and then promptly placed on ice and centrifuged for 10 min at 4 °C at 21,400 × g. The supernatants (13 µL) were analyzed by spectrochromatography upon injection onto a Superdex 200 Increase 5/150 column (Cytiva) run at 0.45 mL min⁻¹. The amplitudes of the maximum UV or visible absorbance for the apex of the protein peak detected on the elution profile were plotted against temperature. The sigmoidal dependences were approximated by Boltzmann equation to derive the half-transition temperature for either curve. This method allowed high measurement reproducibility, yielding melting temperatures consistent with the differential scanning calorimetry data published for BmCBP(ZEA)[40].

## Chitin and chitosan-binding assay

To test the ability of BBP to directly bind to chitin, we used the following assay. First, 2.0–2.1 mg of chitin flakes were weighted in 0.5 mL tubes, resuspended in 20 µL of SEC buffer (20 mM Tris-HCl pH 7.6, 150 mM NaCl, 3 mM βME), and incubated for 10 min at room temperature for chitin hydration. Then, 20 µL of SEC buffer (negative control) or 20 µL of proteins (BBP(βCar), OCP2(CAN) or BSA) on SEC buffer were gently mixed and incubated in a thermoshaker at 25 °C for 12 h at 700 rpm. Additional controls contained only proteins at exactly the same dilution as in the protein-chitin mixtures, and did not contain chitin. After the incubation, the samples were centrifuged for 30 min at 4 °C and 21,400 × g, and we took a picture of their appearance using a camera. The tentative chitin-binding proteins in the pelleted fraction were first washed with 40 µL of SEC buffer and then with 40 µL of SEC buffer additionally containing 2 M urea. Each step included a prolonged incubation stage (8–12 h) at 25 °C and 700 rpm shaking and centrifugation (20 min, 4 °C, 21,400 × g). The last elution was carried out by SDS-PAGE sample buffer to wash out proteins tightly stuck to chitin. Small aliquots of each fraction collected during the assay were subjected to regular SDS-PAGE.

To test the possibility that BBP(βCar) interacts with soluble non-acetylated chitin analog, chitosan, we first dissolved the latter in a minimal volume of 0.1 M acetic acid to protonate chitosan's amino groups and then quickly diluted the transparent starter solution by 10 mM MES pH 6.0 containing 150 mM NaCl. After 30 min incubation at room temperature, the stock chitosan solution was centrifuged but it did not result in any precipitate. We then selected the appropriate chitosan load by obtaining a series of elution profiles by injecting various amounts of chitosan (90–950 µg) on a Superdex 200 Increase 5/150 column (Cytiva) pre-equilibrated into MES buffer indicated above. The flow rate was 0.45 mL min⁻¹. The elution profiles were followed by refractive index changes using an Agilent 1260 Infinity II system equipped with a WR diode-array detector (G7115A, Agilent) and a refractometric detector (G7162A, Agilent, operated at 35 °C). The selected amount of chitosan (90 µg) was then mixed with either MES buffer or two concentrations of BBP(βCar) and analyzed as above by SEC with diode-array and refractometric detection. The delay caused by the consecutive order of detectors connected by a PEEK capillary was subtracted from the RI profile to align it with the profile followed by absorbance. While the absorbance of chitosan at 280 and 450 nm was negligibly small, BBP(βCar) exhibited a rather big RI signal. The possible changes of the BBP elution profile in the presence of chitosan were, therefore, identified by comparing of BBP-specific absorbance in the absence or the presence of chitosan as well as by comparing the overall RI signal for the chitosan-BBP mixtures with the algebraic sum of the individual RI signals for BBP and chitosan. Both changes were very low indicating rather weak interaction between BBP and chitosan.

## Statistics and reproducibility

Spectroscopy and spectrochromatography data were collected at least three times, for each of at least two separately purified protein batches obtained, and the most typical results are presented (no significant outliers were observed).

## Reporting summary

Further information on research design is available in the Nature Portfolio Reporting Summary linked to this article.

## Data availability

The uncropped versions of the gels are shown in Supplementary Fig. 6. The source data behind the graphs in the paper can be found in Supplementary Data 1. All other data are available from the corresponding author upon request.

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

## Acknowledgements

We are thankful to Prof. Thomas Friedrich (TU Berlin) for providing *E. coli* strains synthesizing carotenoids, to Dr. Evgenia Yu. Parshina (Department of Biophysics, Faculty of Biology, Moscow State University) for Raman spectra measurements, to Prof. Valery P. Varlamov (Institute of Bioengineering, FRC of Biotechnology of the Russian Academy of Sciences) for providing chitin and chitosan preparations, to Yury B. Slonimskiy for help with BBP construct design and to Dr. Olga S. Korsunovskaya (Entomology Department, Faculty of Biology, Lomonosov MSU) for providing the two locust specimens. The study was partly supported by the Ministry of Science and Higher Education of the Russian Federation (075-15-2021-1354). SEC-MALS, mass-spectrometry and circular dichroism measurements were done at the Shared-Access Equipment Centre "Industrial Biotechnology" of the Federal Research Center "Fundamentals of Biotechnology" of the Russian Academy of Sciences.

## Author contributions

N.N.S.: initiated and coordinated the study; N.A.E., E.E.D. and N.N.S.: expressed and purified proteins; N.A.E. and N.N.S.: designed and performed experiments; NAE: performed HPLC analysis and worked with locusts; N.A.E., E.G.M., and N.N.S.: analyzed data and discussed the results; N.N.S. wrote the paper with input from N.A.E.; N.N.S. prepared illustrations and graphs with the input from N.A.E.; N.N.S.: supervised the study.

## Competing interests

The authors declare no competing interests.
