## [Peer Review File · Communications Biology]

Reviewers' comments:

Reviewer #1 (Remarks to the Author):

In this paper, the authors reported a technical trial of β CBP expression, binding features of β CBP with carotenes, and molecular interactions of β CBP with insect cuticle components through a series of biological, chemical and physical techniques. Totally, this paper is only a synthesis of technical experiments based on previous techniques, lacking of technical innovation and biological hypothesis.

Minor comments:

1. Abbreviations used in the text: β -carotene-binding protein should use β CBP as used in previous literatures.
2. Line 110-113, The species described in this sentence should be both migratory locusts and desert locusts as refs 27 and 28 indicated, but not only desert locust.

Reviewer #2 (Remarks to the Author):

To Authors,

This paper reported the characterization of carotenoprotein found from the desert locust that binds with β -carotene compared to B.mori carotenoid binding protein and Orange Carotenoid Protein 2 from *Gloeocapsa* sp. PCC 7418. Rapid body color changes have been reported in several kinds of arthropod species. However, the molecular mechanism is not fully understood. This paper shows important findings on the connection between carotenoid transport and body coloration in desert locusts. These findings are valuable for entomologists and researchers in a wide range of scientific fields. Unfortunately, the introduction and discussion section focuses only on molecular properties and lacks a biological perspective and data for β -carotene transfer from BBP to chitin. Yang M. et al. reported that BBP is located in the epidermal cells between the cuticle layer and the basement membrane. Since BBP is not a membrane protein, the authors need clear evidence of how it transports carotenoids to the cuticle layer. Based on your new findings, I think the discussion section needs to be reconstructed with biological insight.

Major comments;

- 1) Since the last part of the discussion describes the mechanism of carotenoid deposition on cuticles, it is easier for the reader to understand if you focus on the background that leads to that claim and introduce. In the introduction section, it is better to describe about body color of animals related to carotenoids and then introduce the role for carotenoid binding proteins and your motivation (introduction section).
- 2) Yang M. et al. reported that BBP is located in the epidermal cells between the cuticle layer and the basement membrane. Since BBP is not a membrane protein, the authors need clear evidence of how it transports carotenoids to the cuticle layer.

If the authors insist the role of BBP and the molecular mechanism of locust cuticle coloration, the authors should show the relationship between chitin sclerotization and coloration period in desert locust (line 126).

- 3) Why did not the authors carry out only β -carotene control in this experiment? (Figure 5). I think the authors need to clear that β -carotene did not bind without carotenoid binding proteins.
- 4) The authors mentioned the protein stability and carotenoid preference of BBP compared to BmCBP and OCP2. However, it is not stated what the benefit of this difference in nature is for the desert locust (Fig. 2 and 3). The authors should mention more how these properties play a role in the coloration mechanisms in the desert locust.
- 5) The authors compare β -carotene signatures in BBP by Raman spectroscopy using the new and old locust cuticle (Fig. 3F) and found that several Raman spectra were different between the new and old desert locust. I think the authors can state that BBP is stable in terms of pigment retention for 64 years. Therefore, the authors need to change the order in which the data are presented to focus on pigmentation in the cuticle.
- 6) The authors insist that the yellow coloration of gregarious locusts is determined by upregulated BBP, its secretion, and reversible binding of the BBP-carotenoid complex as a unit to chitin (lines 452-454). If so, the authors should show a relationship between increased BBP and chitin staining.
- 7) This manuscript becomes more interesting when the authors will add biological insight and data for coloration in desert locust.

Minor comments;

- 1) β -carotene binding protein should put as BBP here (line 131, last line of the introduction).
- 2) BmCBP has been shown to bind alpha and beta carotene and lutein. It has also been shown to have a high ability to bind zeaxanthin. However, it has not been shown not to bind β -carotene (lines 102). Please check following paper; *Biochemistry*. 2011 50(13): 2541–2549. doi:10.1021/bi101906y, and *JBC* 2002 277(35): 32133-32140. doi:10.1074/jbc.M204507200
- 3) It is better to include only the results obtained from your data in the Results section. The discussion is described in the Discussion section. (in the Result section).

Reviewer #3 (Remarks to the Author):

Comments

The manuscript titled “Molecular insights into the mechanism of yellow body coloration of gregarious locusts by stress-tolerant chitin-binding carotenoprotein” deciphers the possible mechanism of body coloration in locusts. The manuscript has presented an interesting topic with through details, however, there are some issues/concerns that deserve further attention from the authors.

General Comments:

1. Lines 180-185; what could be the possible reasons for the difference of the calculated structural model from the actual conformation of BBP? While the research provides detailed insights into the expression, purification, and stability of BBP, the lack of a solved structure limits a comprehensive understanding of its conformation and interactions.
2. Lines 186-189; the change in the chromatographic peaks is obvious and expected. I wonder how the partially purified BBP being a mix of the proteins could show identical peaks. The identification of the

other two small peaks of the native BBP as shown in Fig. 2A needs further attention.

3. Line 383; the statement needs revision, “especially ... available”.

4. Line 417; use abbreviations for CAN and ECH.

5. At many places the sentences are very long and confusing. Revising these sentences for easy understanding and clarity will be appealing to the readers.

We are sincerely thankful to the referees for the valuable comments and recommendations. We note that the serious analysis of these recommendations made us rewrite significant portions of the introduction and discussion sections of our paper, in an attempt to provide more biological perspective and make the paper more interesting to a general biologist. In particular, this resulted in expanding bibliography by adding more relevant references to support new facts and statements concerning insect coloration mechanisms based on carotenoids. We also removed parts of the text on unrelated carotenoproteins to stay more focused. In addition, we tried to eliminate repeats and make longer sentences shorter for more clarity.

We very much hope that the amendments introduced will make the referees happy and supporting publication of our revised paper. Our point-by-point responses to referees comments are found below (in blue).

Reviewers' comments:

Reviewer #1 (Remarks to the Author):

1.1 In this paper, the authors reported a technical trial of β CBP expression, binding features of β CBP with carotenes, and molecular interactions of β CBP with insect cuticle components through a series of biological, chemical and physical techniques. Totally, this paper is only a synthesis of technical experiments based on previous techniques, lacking of technical innovation and biological hypothesis.

We would like to thank this referee for the critical reading of our manuscript. We respectfully disagree with the subjective opinion of this referee on the technical nature of our paper, which is apparently not shared by other two referees. Our study was motivated by the idea that it is the macromolecular complex of BBP with the carotenoid that is the pigment and that BBP is not simply a transporter of carotenoids. This idea could have been confirmed directly by reconstitution of this complex, and we used beta-carotenoid producing *E. coli* cells to successfully do this for the very first time. This was especially important because 'previous techniques', as put by this referee, showed quite contradictory results on many properties of BBP, as justified in the introduction of our paper. The manuscript also contains important new findings, for example, the oligomerization state of BBP in complex with the carotenoid, its carotenoid binding preferences, its spectral signatures (very different from any known for carotenoproteins and for free beta-carotene in different environments), its unparalleled stability to heat and acid/alkali. None of these findings has been reported before nor could be expected from previous, mostly physiological and molecular biology works. The chitin binding assays were solely hypothesis driven. In addition, we introduced here a novel technique for assessing thermal stability of carotenoproteins that, to the best of our knowledge, has not been described before but will be useful for scientists from different fields. In any case, we were motivated by great previous works, which we duly cited and discussed (adding several ones during the revision). We believe that our study nicely cements previously known facts together, indeed providing novel molecular insights into the molecular/biochemical mechanism of the yellow body coloration of gregarious locusts, thereby deserving publication.

Minor comments:

1.2. Abbreviations used in the text: β -carotene-binding protein should use β CBP as used in previous literatures.

We have considered this name seriously but decided to use BBP to avoid simple confusing of β CBP with the CBP from *Bombyx mori* (sometimes referred to as BmCBP) and because in many instances the use of the Greek letter is restricted in the internet and is not stably recognized by search machines. There is also an even more confusing name of the same protein, given long before it was sequenced, namely, the

'yellow protein'. We see this naming very unfortunate and not informative at all. Moreover, there is another factor in locusts related to melanization-associated protein Yellow (YEL). In essence, we prefer to stick to the name BBP, which is the simplest and most unique at least in the subject area. However, we have now acknowledged all alternative names of this protein used in previous literature: YP, YPT, betaCBP.

1.3. Line 110-113, The species described in this sentence should be both migratory locusts and desert locusts as refs 27 and 28 indicated, but not only desert locust.

We gratefully changed this sentence accordingly. Having advised by Reviewer 2, we have re-considered the introduction and discussion of our paper, which we hope resulted in a much better story.

Reviewer #2 (Remarks to the Author):

2.1. To Authors,

This paper reported the characterization of carotenoprotein found from the desert locust that binds with β -carotene compared to B.mori carotenoid binding protein and Orange Carotenoid Protein 2 from *Gloeocapsa* sp. PCC 7418. Rapid body color changes have been reported in several kinds of arthropod species. However, the molecular mechanism is not fully understood. This paper shows important findings on the connection between carotenoid transport and body coloration in desert locusts. These findings are valuable for entomologists and researchers in a wide range of scientific fields.

Unfortunately, the introduction and discussion section focuses only on molecular properties and lacks a biological perspective and data for β -carotene transfer from BBP to chitin. Yang M. et al. reported that BBP is located in the epidermal cells between the cuticle layer and the basement membrane. Since BBP is not a membrane protein, the authors need clear evidence of how it transports carotenoids to the cuticle layer.

Based on your new findings, I think the discussion section needs to be reconstructed with biological insight.

We are delighted to receive such positive reference and are thankful to this reviewer for the kind evaluation of our manuscript. Moreover, we are particularly glad that his/her recommendation helped us realize that we indeed need to reshape the manuscript in order to make it more appealing for a general biological audience. Deep literature search and analysis helped us add and discuss several relevant works that were previously omitted. We completely restructured Introduction by removing all less important and too much detailed facts and adding new aspects of coloration in arthropods and insects in particular. We significantly rewrote discussion by removing most of repeats and providing more arguments towards the proposed locust cuticle coloration mechanism (Fig. 5). We also toned down some our statements by outlining that "Our study provides the detailed characterization of BBP and thereby complements previous works in explaining the remarkable molecular mechanism of locust cuticle coloration" and giving an additional tribute to previous works. We very much hope that the revised version of our paper will be welcomed by the referee.

Major comments;

2.2) Since the last part of the discussion describes the mechanism of carotenoid deposition on cuticles, it is easier for the reader to understand if you focus on the background that leads to that claim and introduce. In the introduction section, it is better to describe about body color of animals related to carotenoids and then introduce the role for carotenoid binding proteins and your motivation (introduction section).

We have changed the introduction accordingly, by removing all unnecessary details and focusing on animal coloration mechanisms based on carotenoids and adding more information on the coloration and biology of locusts.

2.3) Yang M. et al. reported that BBP is located in the epidermal cells between the cuticle layer and the basement membrane. Since BBP is not a membrane protein, the authors need clear evidence of how it transports carotenoids to the cuticle layer.

If the authors insist the role of BBP and the molecular mechanism of locust cuticle coloration, the authors should show the relationship between chitin sclerotization and coloration period in desert locust (line 126).

We have changed the introduction significantly and hope that the stage is now set more properly. Analysis of rich literature on physiology and ethology of locusts, molecular biology results of Prof Le Kang laboratory and our biochemical and spectroscopic data allowed us to more confidently conclude that the yellow pigment responsible for the locust cuticle coloration is not beta-carotene itself, but its complex with BBP. In other words, this protein is not functioning as a shuttle and is a terminal depot for beta-carotene and is therefore a macromolecular pigment (similar to beta-crustacyanins from lobster shell and blue carotenoproteins from sea sponge). First, the protein is easily extracted from the cuticle and comes up fully loaded with beta-carotene (no significant amount of apoprotein is present). Second, the apoform of BBP could not be obtained even recombinantly, indicating that it is rather unstable. Third, the holoform BBP-carotenoid is very stable to various stresses, which preserves the chromophore in a state required for the coloration. Fourth, Raman spectra indicate that BBP in vitro and in situ have very similar characteristics and that these characteristics are different from those of free beta-carotene in solution. Fifth, BBP is predicted as an extracellular protein and is found in mature functional form with already removed N-terminal signal peptide, indicating that its destination is pigment granules and then, extracellular space. We also added in discussion and in an altered Fig. 5 that the color is likely contributed by BBP-bCar complexes in the cuticles and in the intracellular pigment granules. This scenario is very well in agreement with the immunogold staining (Le Kang) and that BBP-bCar is extracted from cuticles and the attached epidermis (Wybrandt).

2.4) Why did not the authors carry out only β -carotene control in this experiment? (Figure 5). I think the authors need to clear that β -carotene did not bind without carotenoid binding proteins.

Thank you for this remarkable question. Unfortunately, this control will not be informative and, moreover, it would be misleading due to the reasons we carefully considered and discussed in the revised version of our paper. The added passage is: "Importantly, we did not use β -carotene as a control because it would in any case immediately precipitate in aqueous buffer and would therefore irreversibly stain the chitin pellet, not providing any relevant information." In other words, the essence of the method we used was to separate soluble and insoluble fractions, and bCar will always go to the latter no matter what. Please also see our other arguments above against the fact that the pigment is bCar itself and not its complex with BBP (Raman spectra, stability and full loading of BBP by bCar, etc).

2.5) The authors mentioned the protein stability and carotenoid preference of BBP compared to BmCBP and OCP2. However, it is not stated what the benefit of this difference in nature is for the desert locust (Fig. 2 and 3). The authors should mention more how these properties play a role in the coloration mechanisms in the desert locust.

Thank you for the great question. Indeed, it was our omission. We added the following phrase to discussion: "The remarkable heat and chemical stability of BBP in complex with the abundantly available β -carotene can be beneficial for the pigment preservation under the harsh environment conditions potentially experienced by the desert locusts, including desert heat and sunlight exposure, which would be damaging for the unbound β -carotene in the absence of BBP."

2.6) The authors compare β -carotene signatures in BBP by Raman spectroscopy using the new and old locust cuticle (Fig. 3F) and found that several Raman spectra were different between the new and old desert locust. I think the authors can state that BBP is stable in terms of pigment retention for 64 years. Therefore, the authors need to change the order in which the data are presented to focus on pigmentation in the cuticle.

Following this insightful recommendation, we have now reconsidered all storytelling and placed Raman spectra with the locusts in Fig. 1. Next we introduce the extracted native BBP and its recombinant counterpart, show their equivalence and only then switch to a more in depth BBP characterization. We agree that this order is much more logical and easy to follow for a general reader. Thank you again.

The new Fig. 1:

Panels a-c went from former Fig. 3 to meet the recommendation of the reviewer. Accordingly, panels d and e show the comparison of native and recombinant BBP and partially went from former Fig. 2. Former panels b and c showing spectrochromatogram data as heatmaps were discarded as the same content is shown in new Fig. 1h.

The new Fig. 2.:

Panel a b and c were formerly c a and b, changed according to the changes in the text.

The new Fig. 3.: (part of the data with Raman spectra went to the new Fig. 1)

The new Fig. 5.:

Panel e is redrawn for better clarity.

2.7) The authors insist that the yellow coloration of gregarious locusts is determined by upregulated BBP, its secretion, and reversible binding of the BBP-carotenoid complex as a unit to chitin (lines 452-454). If so, the authors should show a relationship between increased BBP and chitin staining.

Please see above. We seriously analyzed literature and added new relevant references, which we believe unequivocally define BBP-βCar complexes as the authentic macromolecular pigment located

both in the pigment vesicles in the epidermal cells and in the cuticle (extracellularly). The dose-dependences and the upregulation of BBP transcription and translation were studied earlier (2018, 2022, 2023), and all this literature is duly cited in the revised version.

2.8) This manuscript becomes more interesting when the authors will add biological insight and data for coloration in desert locust.

Please also see above. We thankfully agree with this point and would like to say that it made us completely reconsider our paper and hopefully improved it substantially. The following passage was introduced (introduction section): “A very peculiar case of insect coloration is represented by locusts - important and abundant pests that quickly eat and breed, switching from solitary forms to swarms and seriously threatening agriculture by rapid extermination of plantations. Therefore, locust physiology is in focus of intense research aimed at eventually controlling the locust gregarization and preventing substantial economic losses [25]. Locusts are known for their polyphenism associated with the ability to change body color at different phases of their development and lifestyle, from green to black/brown and bright yellow. They feed on various plant sources and therefore consume a lot of carotenoids [26,27]. While roughly similar carotenoid content is found in females and males of gregarious locusts, only males acquire bright yellow color in high-density populations, i.e., in the gregarious phase, to avoid harassment from other mature males [26–29]. This yellow coloration, also known to be induced in nymphs at high temperatures, is stimulated in males by juvenile hormone and probably some sex hormone(s), whereas melanization is controlled by the special neuropeptide corazonin [30–33]. Yellowing of gregarious locusts, such as *Schistocerca gregaria* and *Locusta migratoria*, results from the male-specific expression of the special carotenoprotein found in cuticles and epidermis [27–30,34,35]. Recently it was found that this is achieved by activation of protein kinase C alpha in response to crowding: PKC α phosphorylates the activation transcription factor 2 at Ser327 to promote its binding to the yellow protein promoter and induce overexpression [36].”

Minor comments;

2.9) β -carotene binding protein should put as BBP here (line 131, last line of the introduction).

In this particular case we meant more a bCar binding protein more generally. To avoid confusion this phrase changed to: “Our work also paves the way to structural studies and potential applications of this unique β -carotene-binding protein.”

2.10) BmCBP has been shown to bind alpha and beta carotene and lutein. It has also been shown to have a high ability to bind zeaxanthin. However, it has not been shown not to bind β -carotene (lines 102). Please check following paper; Biochemistry. 2011 50(13): 2541–2549. doi:10.1021/bi101906y, and JBC 2002 277(35): 32133-32140. doi:10.1074/jbc.M204507200

The corresponding phrase changed to “Besides its native ligand lutein, BmCBP binds various xanthophylls including the lutein’s isomer zeaxanthin, as well as traces of carotenes [21–23].” Where ref 23 is the mentioned paper “H. Tabunoki, H. Sugiyama, Y. Tanaka, H. Fujii, Y. Banno, Z.E. Jouni, M. Kobayashi, R. Sato, H. Maekawa, K. Tsuchida, Isolation, characterization, and cDNA sequence of a carotenoid binding protein from the silk gland of *Bombyx mori* larvae, J Biol Chem 277 (2002) 32133–40. <https://doi.org/10.1074/jbc.M204507200>.”

2.11) It is better to include only the results obtained from your data in the Results section. The discussion is described in the Discussion section. (in the Result section).

Thanks. We made our best to remove any distracting points from the Results section.

Reviewer #3 (Remarks to the Author):

3.1. Comments

The manuscript titled “Molecular insights into the mechanism of yellow body coloration of gregarious locusts by stress-tolerant chitin-binding carotenoprotein” deciphers the possible mechanism of body coloration in locusts. The manuscript has presented an interesting topic with through details, however, there are some issues/concerns that deserve further attention from the authors.

We are grateful to this reviewer for the kind evaluation of our work. The recommendations were taken seriously and the corresponding changes are adopted in the revised version.

General Comments:

3.2. Lines 180-185; what could be the possible reasons for the difference of the calculated structural model from the actual conformation of BBP? While the research provides detailed insights into the expression, purification, and stability of BBP, the lack of a solved structure limits a comprehensive understanding of its conformation and interactions.

Thank you very much for the good question. We agree that the structural data would be a saint Graal in this story, but as we put in the revision, all our attempts to get crystal quality crystals of BBP have so far been unsuccessful. In fact, we managed to crystallize the protein but the best diffraction so far was about 6 Å. We added in the revision that the mentioned difference in the CD spectrum may stem from the conformational reorganization upon carotenoid binding.

3.3. Lines 186-189; the change in the chromatographic peaks is obvious and expected. I wonder how the partially purified BBP being a mix of the proteins could show identical peaks. The identification of the other two small peaks of the native BBP as shown in Fig. 2A needs further attention.

We respectfully disagree that it was obvious that the recombinant and native BBP would be similar. We believe this is one of the key experiments described in our study – that we compared the reconstituted and native BBP extracted directly from cuticles. It was not a given right to expect that those would be similar and we are happy that it turned like this. Also, our statement was slightly different. We meant that the larger absorbance at 280 nm would be explained by the presence of other proteins which are normally present in crude extracts, including the POI apoform. The presence of the second peak at V_0 in native BBP likely indicates the presence of either soluble aggregates on membrane fraction colored due to the integration of residual carotenoids, and this is very often seen in rough samples of carotenoproteins extracted from natural sources.

3.4. Line 383; the statement needs revision, “especially ... available”.

We removed this sentence.

3.5. Line 417; use abbreviations for CAN and ECH.

We added these abbreviations.

3.6. At many places the sentences are very long and confusing. Revising these sentences for easy understanding and clarity will be appealing to the readers.

We significantly edited our text by splitting long sentences and simplifying the phrases. Thank you!

REVIEWERS' COMMENTS:

Reviewer #1 (Remarks to the Author):

This version of manuscript has improved a lot. I have no more questions.

Reviewer #2 (Remarks to the Author):

Thank you for your response to my opinion. This manuscript has been revised than before.

Reviewer #3 (Remarks to the Author):

Besides addressing the reviewer's concerns satisfactorily, the authors have made substantial improvements to the revised manuscript. I recommend the revision for possible publication.

We would like to thank the reviewers for supporting publication of the revised paper. No detailed responses are necessary.

REVIEWERS' COMMENTS:

Reviewer #1 (Remarks to the Author):

This version of manuscript has improved a lot. I have no more questions.

Reviewer #2 (Remarks to the Author):

Thank you for your response to my opinion. This manuscript has been revised than before.

Reviewer #3 (Remarks to the Author):

Besides addressing the reviewer's concerns satisfactorily, the authors have made substantial improvements to the revised manuscript. I recommend the revision for possible publication.